# OpenReview forum: "The Paradox of Robustness: Decoupling Rule-Based Logic from Affective Noise in High-Stakes Decision-Making"
_TMLR — Rejected by TMLR_

### Review · Reviewer_rsr6 · 2026-02-27

**Summary Of Contributions:**

This paper tests the hypothesis whether affective narratives, such as claiming hardships in loan underwriting decision making, influences the outcome when an LLM is explicitly instructed to output  the decision based on fixed and deterministic rules. The study does not find any statistically significant effect of affective narratives on outcomes across 8 API and open models. This finding is tested by extensive statistical tests and some prompt variations, including explicit instructions to output a given decision regardless of the rules (which models follow).

**Audience:**

No

**Audience Explanation:**

The paper evaluates a specific prompting phenomenon in LLMs, namely effects of affective narratives of LLM decision in rule-based schemas.

Given that there are extremely many different (structured) ways to prompt LLMs in different scenarios, and also an extremely large number of human behaviors documented in psychology literature, there needs to be a justification why this particular phenomenon is worth investigating. Examples are: Prompt rephrasing robustness (questions benchmark validity), Jailbreaks (safety concerns), or Societal Biases (widely accepted relevance and possible harms to demographic groups).

The paper provides two arguments for the present evaluation: (1) Surprise, i.e. LLMs behave differently than humans; and (2) motivates using open models over frontier API models in deployment. Argument (2) is less relevant, because arguably this decision would mostly be informed by cost and performance trade-offs, and robustness to one particular aspect has minor impact in this. (1) is also questionable, because inherently LLMs are not humans, so they may or may not follow human behaviors.

However, there are no or only minor critical implications of the found robustness, so while the investigated question (like all questions) may be interesting to some individuals, it has little significance for the wider community.

**Broader Impact Concerns:**

Broader Impact Statement is sufficient in the current version.

**Claims And Evidence:**

No

**Claims Explanation:**

The paper extensively tests statistical properties of the conducted experiments and hence it is clear that the evidence against for null hypothesis within the evaluated setup is convincing.

However, the overall experimental setup suffers from fundamental limitations:
 1. The benchmark contains only 3 scenarios in 3 variations (9 total) and 6 different affective narratives. While the combination of different components leads to higher effective number of prompts and repeated sampling from LLMs yields higher number of responses, inherently the variability of prompts is limited. Scientific rigor in evaluating LLM properties usually demands a few hundred diverse samples, at least, even if it is likely that the findings in this paper would hold regardless, because of their extreme narrow focus.

 2. Critically, the paper claims the tested scenarios and affective narratives are natural. However, the paper only lists them in the supplementary without any justification why they were chosen and without any explanation how they were created (e.g., taken from previous studies, designed by domain experts, ...). Given that the benchmark is the central part of the paper, a more comprehensive discussion of its construction is necessary.

3. Furthermore, the paper finds no statistically significant effects, but also does not detail what efforts were undertaken to find effects of affective narratives. The paper argues that the setup is realistic (i.e., using json inputs), but since the implied claim is that LLMs are _never_ influenced by affective narratives in rule-based decision making, a more fundamental approach than minor prompt variations would be necessary to find cases where affective narratives actually do have an effect. For example, the paper could apply ORPO [1] or similar prompt evolution strategies and use an LLM as adversary to automatically design cases where affective narratives have an influence on decision (or provide compelling evidence that such cases are extremely hard to find).

Beyond these fundamental problems (benchmark size, construction, and generalization from proposed setup $\rightarrow$ general LLM property), there are also minor issues with writing and experimental setup:
 * The paper claims that using API models is "reproducible", but generally this is not the case, as proprietary providers may change models without even notifying users. In fact the paper itself provides the counterargument: Two preregistered models could not be evaluated because the API that was used for experiments didn't support them any more (p. 17).
 * The paper is very hard to read. In particular, many details and explanations are missing. For example, many abbreviations are never defined, or only after their first use (e.g., RAFR). Furthermore, many of the acronyms denoting statistical tests or quantities may not be known to general readers in the LLM domain and could be better explained. Finally, because the benchmark scenarios and prompting strategies are never explained in the main paper, references to particular scenarios (e.g., "Tversky-Kahneman Asian disease problem, p. 11; paragraph "Contamination control", p. 13) are confusing. Also references to entities like "inadmissible_facts_ignored" (p. 2) are not explained.
  * Design decisions are generally insufficiently motivated. For example, the distinction of frontier LLM models into "US RLHF", "Constitutional AI", and "Chinese Ecosystem" is not explained. What are the definitions of these categories, and what properties distinguish models in them?
  * Some models do not follow prompts or return invalid responses. At least (manually) optimize prompts so models reliably return valid answers. Given that the setup assumes a fixed set of possible decisions, this is possible.

**References**:\
[1] Yang, Chengrun, et al. "Large language models as optimizers." The Twelfth International Conference on Learning Representations. 2023.

**Requested Changes:**

The paper needs significant revision to be recommended for acceptance, mainly:
 1. Expand the benchmark and motivate its construction, so the paper reads less like a case study but rather a broad and thorough empirical evaluation.
 2. Provide stronger arguments that affective narratives _never_ influence LLM decisions in rule-based tasks.
 3. Revise the structure and explanations in the paper to improve clarity. Provide missing explanations and always define acronyms at their first occurrence. I also suggest a dedicated "Tools / Methods" section to briefly explain and motivate the statistical tools used throughout the paper.
 4. Update prompts to eliminate invalid responses from models.

---

> ### Author Response · Authors · 2026-04-04
> **Benchmark Expansion, Adversarial Evidence, and Clarity**
>
> Requested Change 1: Expand the benchmark and motivate its construction.
>
> We addressed this concern in three ways. First, we added a construction methodology paragraph (Section 3.4) documenting domain selection criteria, scenario design process, ground-truth verification, and narrative construction methodology, with a new Table 2 providing a 9-row scenario overview. Second, we conducted a reviewer-driven immigration extension as a fourth-domain boundary probe (Section 4.4). Five scenarios spanning family sponsorship, asylum timeliness, naturalization physical presence, TPS registration, and H-1B wage compliance were selected from a pool of ten candidate rule sets using a six-dimension scoring rubric (threshold determinism, emotional salience, evidence-affect separation, regulatory grounding, benchmark diversity, and adversarial narrative potential), followed by a second-stage legal and construct-validity review. The extension yielded 14,183 valid responses with Δ=+0.8% (within the ±3% ROPE). Third, we added a compact running example in Section 3.2 showing how Conditions A/N/E work for a concrete scenario (F1 mortgage refinance), so the benchmark logic is immediately legible without consulting the appendix.
>
> We respectfully clarify the experimental scale. While the core benchmark comprises 9 base scenarios, each generates 18 condition variants (162 unique prompts), and the paper reports 84,245 valid responses across 13 experiments. The I^2=0.0% heterogeneity confirms consistency across scenarios.
>
> Requested Change 2: Provide stronger arguments that narratives never influence decisions.
>
> We conducted two new analyses. First, an adversarial narrative pilot (Section 4.4, Table 7) using LLM-generated maximally persuasive narratives across three strategies (emotional maximization, implicit evidence embedding, authority/social pressure). The aggregate adversarial shift is Δ=+0.6% (95% CI [−3.6%,+4.8%]), within ROPE. This is approximately 85x smaller than the instruction override effect (+51.0 pp), confirming the dissociation between narrative robustness and instruction hierarchy robustness. Second, the immigration extension provides a particularly stringent test: the asylum scenario, which features torture and persecution narratives (the most emotionally extreme content in the benchmark), produces exactly zero outcome shift across all eight models.
>
> We note the reviewer's OPRO reference (Yang et al. 2023) addresses task performance optimization; our adversarial protocol is designed specifically for vulnerability discovery in an already-solved task.
>
> Requested Change 3: Improve clarity.
>
> All acronyms are now defined at first use (RAFR, BCa, GEE, ROPE, MDE, CFPB, ESI, FERPA). A new statistical framework paragraph (Section 3.3) provides a self-contained primer for ML audiences. A running example (Section 3.2) concretely illustrates Conditions A/N/E. The model taxonomy (US RLHF, Constitutional AI, Chinese ecosystem, Open-source RLHF) is defined in Section 3.5, with Bai et al. (2022) cited for the Constitutional AI category specifically.
>
> Requested Change 4: Update prompts for invalid responses.
>
> We intentionally use identical prompts across all models to maintain a controlled comparison; per-model prompt optimization would confound prompt engineering quality with narrative robustness measurement. This design choice is now explicitly defended in Section 3.5. Attrition is 4.1% overall and non-differential for 7/8 models; frontier models achieve 0% attrition.
>
> On audience interest (Criterion 2):
>
> The EU AI Act classifies AI in creditworthiness assessment and certain healthcare and educational applications as high-risk, requiring conformity assessments that include bias testing---domains closely related to those in our benchmark. Concurrent work shows that source framing (Germani & Spitale, 2025) and moral framing (Cheung et al., 2025) do shift LLM behavior, while our finding that affective narrative framing does not identifies a non-obvious boundary. TMLR's Criterion 2 guidelines state: "A reviewer should always assume that at least some individuals would be interested in any paper, unless the reviewer is confident that this is not the case." We believe the regulatory, comparative, and methodological contributions clear this threshold.

---

### Review · Reviewer_st38 · 2026-03-19

**Summary Of Contributions:**

The submission evaluates whether large language models are sensitive to affective narrative framing in rule-bound decision making. Past research has documented that humans show strong framing effects in various decision making domains. Do LLMs show the same? Previous and concurrent work has documented that LLMs are sensitive to minor prompt perturbations and display sycophantic alignment. In contrast, here the authors show a surprising robustness to affective framing, specifically in decisions requiring rule-based logical decision making. The paper includes carefully designed experiments and a benchmark to characterize and quantify this insensitivity to prompt framing.

Strengths
* The paper is well-written and very thorough and the experiments are well-designed to carefully   test specific hypotheses.
* Limitations are throughly described.
* Careful application of statistical methods to quantify their null result.
* Evaluation of several LLMs differing on various attributes.


Weaknesses
* The authors emphasize the comparison between the large framing effects found in humans and the null effect reported here in LLMs. However, the authors have not conducted a human experiment to quantify framing effects for the specific prompts used here and they provide few references detailing human framing effects in similar task settings, instead focusing of framing effects more broadly.

**Audience:**

Yes

**Audience Explanation:**

This work will be of interest to anyone concerned with the conditions under which LLMs show susceptibility/robustness to prompt framing, which is highly relevant for safe and responsible AI deployment. This work will also be of interest to people interested in comparing human and AI cognition.

**Broader Impact Concerns:**

The authors have more than adequately addressed several potential issues in their Broader Impact Statement.

**Claims And Evidence:**

Yes

**Claims Explanation:**

I am generally convinced of the evidence for absence presented here, although I note the challenge of defending a null result. The experiments are well-designed and appropriate statistics were used. In my requested changes, I challenge the authors to clarify specific claims about the comparison to human effect sizes and the generality of this result for all "instruction-tuned" models, but overall I take the main robustness claim to be supported.

**Requested Changes:**

- It is not clear exactly where the reported human effect size range comes from. The authors state that human framing effects in decision-making typically show Cohen's h = 0.3–0.8 across domains and Figure 4 compares the human and llm effect sizes. Three citations are provided to support this claim. One is the popular book _Thinking Fast and Slow_ by Daniel Kahneman. If this range is reported in that book, please also cite a primary source. One is a theoretical paper about the role of affect in decision making. The other is a classic Kahneman paper about framing effects. These two papers contain various choice proportions, from which one can calculate a Cohen's h, but they do not themselves report a typical effect size range. These articles also include effect sizes outside of the range reported. It is not clear whether the authors calculated this range themselves based on the literature (in which case more details should be provided) or whether it is reported in an existing review article, in which case an additional reference should be included. I'm also not clear what is being plotted in Figure 4. Is this meant to be schematic, or is this actual data? Please include in the caption what the error bars represent. In any case, this figure does not add much and I recommend cutting it.

- Many human framing effects in decision making have been studied in tasks requiring value judgements of risky choices e.g., evaluating whether to accept a specific gamble. Framing can affect risk-aversion and may be used as a shortcut especially when computing the correct response is costly. This is quite different from the rule-bound narratives studied here where the task is to decide whether certain criteria have been met, the decisions don't involve any risk, and may not be especially cognitively demanding. Given the narrower focus of your work on affective framing effects in rule-based decision making, I suggest you attempt to quantify the magnitude of human framing effects in similar rule-bound decision making, not framing effects more generally or in risky choice behaviour. The paper cited later by Steblay et al. (2006) seems especially relevant here.

- The caption for Figure 2 refers to 95% confidence intervals but there are no error bars on the barplot. Please include the error bars for the CIs which are described in the caption to span zero.

Minor suggestion (optional)

- This is partly a minor comment about terminology but also a question about the generality of one of your claims. You refer to the models you test as "instruction-tuned" and conclude that the insensitivity you document is a general property of "instruction-tuned" models. If I am not mistaken, all the models you analyse have undergone an alignment process beyond supervised finetuning, involving preference tuning or other feedback. So we don't know whether supervised fine tuning (the minimal criteria to call a model instruction-tuned) is sufficient to achieve this robustness or whether it emerges in later post-training stages. In revision, you could considered referring to "aligned" instead of "instruction-tuned" LLMs, and you may wish you mention that the current experiments only show that pretrained models can't do the task and that fully aligned models can do so robustly. You don't know which step in post-training is responsible for this robustness (although you could speculate that SFT is enough). You could also consider running your analysis on models that have only undergone SFT to verify.

---

> ### Author Response · Authors · 2026-04-04
> **Human Baseline, Figure Fixes, and Terminology**
>
> Comment 1: Human effect-size sourcing and Figure 4.
>
> The h=0.3-0.8 range is now properly sourced with a new derivation table (Appendix D.4) that traces each bound to a specific meta-analysis: Kuhberger (1998, 136 studies, d=0.31), Pinon & Gambara (2005, 230 studies, d=0.31), Steblay et al. (2006, 48 juror studies, h≈0.30-0.45), and Tversky & Kahneman (1981, h≈1.03). Per the reviewer's recommendation, Figure 4 has been removed; the comparison is now in text (Section 5.1) with Steblay promoted as the primary comparator: "human jurors instructed to disregard inadmissible evidence still exhibit significant framing effects (h≈0.30-0.45)---the closest human analog to our experimental paradigm."
>
> Comment 2: Figure 2 confidence intervals.
>
> Fixed. Figure 2 now includes 95% bootstrap CIs. All CIs span zero, confirming no dose-response relationship across affective tiers.
>
> Comment 3: "Instruction-tuned" vs "aligned" terminology.
>
> We adopted "aligned" throughout (12 of 16 occurrences changed; 4 retained in appendix sections describing the specific SFT training step). A new sentence in Section 3.5 notes: "All models tested have undergone preference optimization (RLHF, RLAIF, or DPO) beyond supervised fine-tuning; we cannot determine which post-training stage is responsible." The SFT-only limitation is acknowledged in Section 6.

---

### Review · Reviewer_egC6 · 2026-03-20

**Summary Of Contributions:**

The authors study the instruction-tuned LLMs' robustness in consequential / rule-bound decision-making. They apply a controlled perturbation framework and find near-total invariance to the changes. Their findings show that LLMs can effectively decouple logical rule-adherence from persuasive affective noise.

Key strengths:
- The paper addresses a well-motivated and important question at the intersection of robustness, alignment, and decision-making under uncertainty.
- The empirical finding of near-zero narrative sensitivity across models and domains is surprising and potentially impactful, especially for high-stakes applications.
- The paper includes extensive ablations and robustness checks, which strengthen confidence in the reported phenomenon.


Key weaknesses:
- The perturbation conditions are mostly deterministic and rule-based, as well as relying on structured prompts, which may make invariance to narrative perturbations expected rather than indicative of a deeper robustness property.
- The authors define narrative vulnerability as a systematic decision shift caused by emotionally-charged but procedurally irrelevant content in rule-bound institutional contexts. While this definition is clear within the experimental framework, it may not fully align with real-world practice, where narrative information can sometimes provide additional context or implicitly relevant signals.
- The study focuses on binary decision outputs and does not analyze internal reasoning processes, making it unclear whether affective information is ignored or internally processed but later discarded.

**Additional Comments:**

N/A

**Audience:**

Yes

**Audience Explanation:**

I believe the findings would still be of interest to the TMLR audience, particularly researchers working on LLM robustness, alignment, and evaluation. The paper addresses an important and underexplored question: whether LLMs inherit human-like susceptibility to emotional framing in decision-making contexts. The reported “Paradox of Robustness” offers a potentially valuable perspective on how instruction-tuned models behave differently across task regimes (e.g., open-ended generation vs. rule-based classification). This distinction has implications for the deployment of LLMs in high-stakes domains such as healthcare, finance, and education. Even if some of the conclusions are contingent on the experimental setup, the benchmark and methodology introduced in this work provide a useful foundation for future studies on robustness to semantic or affective perturbations.

**Broader Impact Concerns:**

The broader impact is discussed in the paper.

**Claims And Evidence:**

No

**Claims Explanation:**

The paper presents a carefully controlled experimental setup, and the results are internally consistent within this framework. However, the evidence does not fully support the broader claims. The benchmark appears overly constrained: tasks are largely deterministic and rule-based, with decisions governed by explicit thresholds; the authors use structured prompts with explicit rules and schema constraints. In such settings, invariance to procedurally irrelevant narrative content is expected, and the near-zero effect size may reflect the rigidity of the task rather than a general robustness property.

Additionally, the internal reasoning process underlying model decisions is not examined in detail. While the final binary outputs appear stable, it remains unclear whether affective information influences intermediate reasoning steps. Such effects may not manifest in the final decision but could still subtly shape the model’s internal decision-making process or confidence. Without analyzing these internal dynamics, it is difficult to conclude whether affective information is truly ignored or instead processed and subsequently overridden at the output stage.

Overall, my main concern is that the benchmark may be too constrained to support the broader robustness claims.

**Requested Changes:**

1. **Clarify the scope of the robustness claim**. The paper should more explicitly acknowledge that the observed invariance may be strongly tied to deterministic, rule-based decision settings. The current framing risks overgeneralizing the result as a broader robustness property of LLMs.
2. **Strengthen discussion of task limitations and construct validity**. The assumption that narrative content is “procedurally irrelevant” is imposed by the experimental design. The authors should more clearly discuss how this assumption differs from real-world scenarios where narrative may contain implicit or contextually relevant information.
3. **Disentangle model robustness from prompt scaffolding**. While ablations are provided, the paper would benefit from a clearer analysis of how much of the robustness arises from structured prompting (rules, schema, role instructions) versus intrinsic model behavior.
4. **Analyze the reasoning traces**. Even if the final output remains unchanged, emotionally charged content may still affect latent deliberation, confidence, or the path the model takes to arrive at its decision. A more detailed analysis of reasoning traces, intermediate justifications, or other process-level signals would help clarify whether affective information is truly ignored or merely overridden at the final output stage.

---

> ### Author Response · Authors · 2026-04-04
> **Scope, Construct Validity, Scaffolding, and Reasoning Traces**
>
> Requested Change 1: Clarify the scope of the robustness claim.
>
> The original framing risked overgeneralization, and we have revised accordingly. The paper now explicitly scopes all claims to rule-bound institutional tasks with explicit correctness criteria:
>
> Abstract: qualified with "in rule-bound institutional decision-making" (revised wording: "strong robustness" rather than "near-total invariance")
> New paragraph in Section 1.2: acknowledges the finding is content-type-specific and task-structure-dependent, citing concurrent work showing other framing types do shift LLM behavior
> Section 5 Discussion: opens with "within the core domain of rule-bound institutional tasks"
> v9 further softens residual overclaims: "field evidence consistent with instruction hierarchy theory" rather than "first empirical validation"
> Requested Change 2: Discuss task limitations and construct validity.
>
> This concern changed the paper materially. A new construct-validity paragraph in Section 1.2 acknowledges that narrative may contain implicitly relevant signals and explains how our design deliberately separates pure affective content from information-bearing content. The construct-validity ablation (Table 5) confirms robustness when narrative is interleaved within the facts JSON as an applicant_statement field (+2.3%, CI spanning zero). The immigration extension pushed this further: in legal domains with waiver or exception pathways, emotionally extreme narratives can imply admissible exception claims, making evidence-affect separation harder to maintain cleanly. We now discuss this openly in Limitations (Section 6).
>
> Requested Change 3: Disentangle robustness from prompt scaffolding.
>
> The scaffolding concern motivated an expanded ablation suite (Section 4.4). Starting from the primary experiment (Δ=−0.1%), we progressively removed scaffolding: removing ignore instructions (+1.0%), removing inadmissibility rules (+0.9%), interleaving narrative within facts JSON (+2.3%), and combining both removals (+2.4%), all with CIs spanning zero. In v9, we present this as "substantially narrowing" rather than "definitively addressing" the scaffolding concern, which we believe is the honest interpretation.
>
> Requested Change 4: Analyze reasoning traces.
>
> A new reasoning-trace analysis examines 720 chain-of-thought outputs from two reasoning models (Section 4.3). DeepSeek R1 (360 traces): 100% of affect-condition traces mention narrative content, and 100% explicitly reject it before applying decision rules---a consistent process-then-override pattern. For o3-mini (360 cells): reasoning token counts show negligible condition difference (d=+0.11). We are careful to note in the paper that this is process evidence, not full mechanistic identification, since neither model gives access to hidden states.

---

### Author Response · Authors · 2026-04-04
**Author Response: Revision Summary and New Experimental Evidence**

We thank all three reviewers for their careful reading and constructive feedback. The revision addresses every requested change with a combination of writing improvements, new analyses, and two reviewer-driven side studies.

Specifically: (1) all robustness claims are now explicitly scoped to rule-bound institutional tasks with explicit correctness criteria; (2) two new side studies (a five-scenario immigration extension with 14,183 valid cells, Δ=+0.8%, within the ±3% ROPE; and a screening-level adversarial narrative pilot with 2,054 cells, Δ=+0.6%, within ROPE) provide additional evidence and boundary-condition analysis; (3) reasoning-trace analysis of two chain-of-thought models reveals a consistent process-then-override pattern; and (4) the human effect-size comparison is now properly sourced with a meta-analytic derivation table.

The revised paper makes a scoped claim: strong robustness in rule-bound institutional tasks, with explicit boundary conditions, not universal immunity to framing. The immigration extension illustrates this scope: it is broadly supportive while surfacing that legally regulated domains with exception pathways can produce small, practically negligible but detectable deviations.

On Criterion 2 (audience interest), the EU AI Act (entering enforcement 2025--2026) classifies AI in creditworthiness assessment and certain healthcare and educational applications as high-risk, requiring conformity assessments that include bias testing---domains closely related to those in our benchmark. Concurrent work demonstrates that source framing (Germani & Spitale, 2025) and moral framing (Cheung et al., 2025) do shift LLM behavior, making our boundary finding (affective framing does not) scientifically informative, not merely a null result.

---

### Author Response · Authors · 2026-04-04
**SUMMARY OF CHANGES**

1. Claims scoped to rule-bound institutional tasks — Abstract, Sec 1.1, 1.2, 5, 7 [egC6, rsr6]
2. Steblay juror comparison promoted — Sec 1, 5.1 [egC6, st38]
3. Construct validity paragraph (implicit evidence) — Sec 1.2 [egC6]
4;. Statistical framework primer — Sec 3.3 [rsr6]
5. Running example (F1 Conditions A/N/E) — Sec 3.2 [rsr6]
6. Construction methodology + Table 2 — Sec 3.4 [rsr6]
7. Model taxonomy + Bai citation — Sec 3.5 [rsr6]
8. Identical-prompt design defense — Sec 3.5 [rsr6]
9. "instruction-tuned" changed to "aligned" — Throughout [st38]
10. "reproducible" changed to "replicable" — Throughout [rsr6]
11. Reasoning-trace analysis (720 cells) — Sec 4.3 [egC6]
12. Scaffolding decomposition + figure — Sec 4.4, Figure 12 [egC6]
13. Adversarial narrative pilot (2,054 cells) — Sec 4.4, Table 7 [rsr6]
14. Immigration domain extension (14,183 cells) — Sec 4.4 [rsr6, egC6]
15. Core-benchmark / extension labeling — Tables 1, 2, 6 captions [All]
16. Asylum zero-shift highlight — Sec 4.4 immigration paragraph [egC6]
17. h = 0.3-0.8 derivation table — Appendix D.4 [st38]
18. Figure 4 removed — Sec 5.1 [st38]
19. Figure 2 tier bars CIs added — Figure 2 [st38]
20. Overclaim softening (8 edits in v9) — Throughout [All]
21. SFT-only limitation — Sec 6 [st38]
22. Waiver-channel design insight — Sec 6 [egC6]
23. Adversarial pilot scope caveats — Sec 6 [rsr6]

---

### Decision · Action_Editor_XBiD · 2026-04-23

**Recommendation:** Reject

**Audience:**

Yes

**Audience Explanation:**

Researchers who are interested in LLM-as-a-judge or bias in AI will be interested in this paper, as it conducted experiments to show that LLM is more robust on emotional framing effects.

**Claims And Evidence:**

No

**Claims Explanation:**

This paper investigates an interesting behavior divergence between LLMs and humans, that is, LLMs appear more robust against affective noise than humans. However, after carefully reading all reviews, the author's response, the revised manuscript, and reviewers' discussions, I tend to reject this paper from a rigorous perspective, as the manuscript still has three main issues that cast doubt on the correctness and generalizability of the article's conclusions:

- **The empirical setup suffers from limitations in both scale and construction.**

    1. As pointed out by rsr6, the constructed dataset scale and scenario numbers are not enough to support their strong claim, leading to concerns in st38 and rsr6.

    2. The dataset construction is also heuristic and lacks detailed descriptions or domain expert analysis, which also limits its claim's correctness, scope, and impact, as all the reviewers admit.

    3. The experiment setup is also not convincing, as pointed out by egC6. I also think it's a deterministic step, and prompt setting may make their finding not surprising because of instruction following or prompt scaffolding. Although their authors added additional results about scaffolding, it is still not enough. Furthermore, as scaffolding is progressively removed, the observed effect size increases monotonically from Δ=−0.1% to Δ=+2.4%, which may also suggest that scaffolding is actively suppressing the effect, reducing their findings reliability and universality, which are also reviewers concerns.

- **Lack of convincing analysis on its reasons**

    As pointed out by st38, the human baseline and their LLM experiments are not structurally aligned, making the central contrast doubtful. In their rebuttal, the authors acknowledge this gap and promote a work by Steblay et al. as the closest human results. However, this study yields a human effect size of only h≈0.30-0.45, the lower end of the cited range, which considerably reduces the claimed human-LLM gap compared to what the abstract suggests. Since conducting a direct human experiment is difficult, I suggest the authors instead apply human-comparable settings to their LLM experiments in future work, which would also address rsr6's concerns about experimental setup.

- **The comparison to human baselines is not sufficiently rigorous to support the paper's central contrast**

    As pointed out by st38, the human baseline and their LLM experiments are not aligned, making its central claim doubtful. In rebuttal, they admit the difference, and the work they cited with a similar setting that also shows similar behavior with their LLM's, reducing the reliability and universality. Since the human evaluation is not easy, I suggest that the authors use humans' settings on LLMs in their future work. It can also address rsr6's concerns about experiment settings.

Due to the three weaknesses, I believe the paper's central claim is not rigorous enough for TMLR. After the discussion, both positive reviewers acknowledged the paper's limitations. Neither felt strongly enough to champion the paper. Therefore, my decision is to reject.